# Radial Growth Adaptability to Drought in Different Age Groups of *Picea schrenkiana* Fisch. & C.A. Mey in the Tianshan Mountains of Northwestern China

**Liang Jiao** *  , **Xiaoping Liu, Shengjie Wang and Ke Chen**

College of Geography and Environment Science, Northwest Normal University, No.967, Anning East Road, Lanzhou 730070, China; lxiaoping18@163.com (X.L.); geowangshengjie@163.com (S.W.); yyckwt@163.com (K.C.)
* Correspondence: jiaoliang@nwnu.edu.cn; Tel.: +86-139-1935-0195

**Abstract:** Forest ecosystems are strongly impacted by extreme climate, and the age effects of radial growth under drought can provide profound understanding of the adaptation strategy of a tree species to climate change. Schrenk spruce (*Picea schrenkiana* Fisch. & C.A. Mey) trees of three age groups (young, middle-aged, and old) were collected to establish the tree-ring width chronologies in the eastern Tianshan Mountains of northwestern China. Meanwhile, we analyzed and compared the response and resistance disparities of radial growth to drought in trees of different age groups. The results showed that (1) drought stress caused by increasing temperatures was the main factor limiting the radial growth of Schrenk spruce, (2) the old and young trees were more susceptible to drought stress than the middle-aged trees, as suggested by the responses of Schrenk spruce trees and based on the SPEI (standardized precipitation evapotranspiration index), and (3) the difference of the resistance indexes (resistance, recovery, resilience, and relative resilience) of three age groups to drought supported that the resistance values were in the order middle age > young age > old age, but the recovery, resilience, and relative resilience values were in the order old age > young age > middle age. These results will provide a basis for the ecological restoration and scientific management of dominant coniferous tree species of different age groups in the sub-alpine forest ecosystems of the arid regions under climate change scenarios.

**Keywords:** *Picea schrenkiana*; age effect; climate response; drought resistance; Tianshan Mountains

## 1. Introduction

Global warming is a major issue affecting natural ecosystems and human society development and, thus, cannot be ignored [1]. The global average surface temperature rose by approximately 0.85 °C over the past 100 years, and it is predicted to rise further by 0.3–0.7 °C in 2016–2035 [2]. Extreme weather is an important manifestation of global warming [3]. Meanwhile, drought has a significant effect on the structure and function of forest ecosystems by causing tree mortality and radial growth rate decline. This, in turn, leads to the loss of structural stability of forest ecosystems because of the high temperature and drought-induced water stress, carbon starvation, cellular metabolic alterations, forest pest and pathogen outbreaks, and forest fires [4,5]. Semi-arid forests receive significant effects from drought, and the mortality events caused by drought stress continuously increased in Central Asia and the southwestern United States since the 20th century [6,7]. At the same time, tree rings have the advantages of high climate sensitivity, precise dating ability, strong continuity, high (annual) resolution, and wide geographical distribution, making them useful in analyzing the effect of climatic and environmental changes (natural disasters, biological disasters, etc.) on tree growth [8,9]. Therefore, based on tree-ring analysis, the response of tree radial growth to climatic factors and its resistance to

drought stress can clearly indicate the ecological adaptation strategies and future development trends of forest ecosystems under global climate change scenarios.

Drought caused by high temperatures is a key factor affecting tree growth in different species or in the same species growing in different regions [10]. Climate warming caused drought stress in forests with good hydrothermal conditions in the tropical humid regions of the Amazon [11]. Heat stress also decreased tree growth rate and caused structural changes in the forests of the eastern and western United States of America and in a semi-arid forest of Europe's Iberian Peninsula [12,13]. Moreover, the water deficit caused by drought events led to an increase in the tree mortality of a boreal forest in Canada and an undisturbed spruce forest in Russia [14,15]. Similar drought vulnerabilities with susceptibility to water failure in different forest ecosystems were found by studying the drought tolerance of global forests. A tree's resistance to drought can be quantified using drought resistance indexes such as resistance (*RT*), recovery (*RC*), resilience (*RS*), and relative resilience (*RRS*) [16]. However, the differences in drought resistance index characteristics of different trees were due to the differences in their species, density, age, and so on [17–19].

Traditional tree-ring climatology holds that the age effects could interfere with the recording of chronological signal by tree rings [20,21]; therefore, it is necessary to eliminate age effects by detrending the data before reconstructing the historical climate and analyzing the growth–climate relationship based on the tree-ring data [22]. However, the physiological responses in different age groups are significantly different because of the differences in the stomatal opening and closing degree, hydraulic conductivity, and hydraulic pressure of trees in different age groups [23]. Therefore, some scholars recently advocated that the age effect should be taken into account when studying the relationship between tree-ring growth and climate [24,25]. Some tree species exhibit a significant age difference in radial growth and climate response. For example, the young trees of species such as *Pinus tabuliformis* Carr. and Schrenk spruce distributed in Central Asia and *Pinus pinaster* distributed on the European Mediterranean coast are more sensitive to climate change than old trees [26]. In contrast, the old trees of *Picea abies* (L.) Karst. distributed on the Black Sea coast of Eastern Europe are more sensitive to climate change than its young individuals [27]. Therefore, the age effect on growth–climate relationships still has uncertainties at different spatial and temporal scales, and verification studies need to be carried out in a wider space, using longer time scales and more tree species. In addition, the age effect can also produce a nonlinear response to drought stress by changing its resource allocation and hydraulic architecture, showing that the resistances to drought stress are discrepant in different age groups of the same species [28]. The young trees of *Pinus sylvestris* L. in Europe exhibited a more rapid response to drought stress [29], while the young trees *Pinus sylvestris* var. *mongolica* Litv. showed poor resistance relative to its adult trees in Inner Mongolia of China [30]. Therefore, the age effect on drought stress capacity also needs to be further explored.

Northwestern China is located deeply in the hinterland of Eurasia with a vast desert, sparse vegetation, and fragile ecosystems [31]. A large virgin forest area in the Tianshan Mountains plays an important role in the natural ecological environment protection of Central Asia. Owing to the blocking effect of the Tianshan Mountains on the airflow, precipitation is greatly reduced, and the climate is more arid in the eastern regions than in the central and western regions of the Tianshan Mountains. Thus, the eastern region was an ideal region to study the effects of climate change on a sub-alpine forest ecosystem [32,33]. Schrenk spruce growing in the eastern Tianshan Mountains is sensitive to climatic factors and susceptible to extreme arid climates, making it an ideal species for analyzing the relationships between radial growth and climate change [34–36]. However, age may cause a series of physiological changes such as changes in tree growth hormone balance, the ratio of photosynthetic and non-photosynthetic tissues, and material absorption and functioning mechanisms [24,37–39]. Therefore, the researches on the response adaptability and resistance of Schrenk spruce trees of different age groups to drought need to be further improved.

Young trees have longer growth windows and faster photosynthesis rates, while older trees have reduced growth rates and enhanced hydraulic constraints; thus, their limited transpiration would affect

their "climate sensitivity" [40–42]. We hypothesized that age would affect the response of different age groups of Schrenk spruce to climate, and that young and old trees would be more sensitive to drought. Moreover, because the middle-aged tree is in a dominant position in the population and has a stronger ability to compete with resources [43], it would show better drought tolerance when facing drought stress. We proposed to (1) identify the main climatic factors limiting the radial growth of Schrenk spruce, (2) compare the response discrepancies of radial growth to drought among the trees of three different age groups, and (3) analyze the discrepancies of resistance indexes to drought conditions among the trees of three different age groups.

## 2. Materials and Methods

### 2.1. Study Area

The study area was located in the eastern Tianshan Mountains, which have a typical temperate continental climate (Figure 1). The annual temperature, total precipitation, and average relative humidity are 220.33 mm, 2.02 °C, and 55.61%, respectively, based on the meteorological data (obtained from 1958–2012) from the Barkol Meteorological Station (43°36′ north (N), 93°03′ east (E), 1677 m above sea level (a.s.l.)), which was the closest to the sampling points (Figure 2). Schrenk spruce was the dominant tree species of the forest ecosystem, mainly distributed in the shady slope at an elevation of 1400–2700 m in the eastern Tianshan Mountains [21].

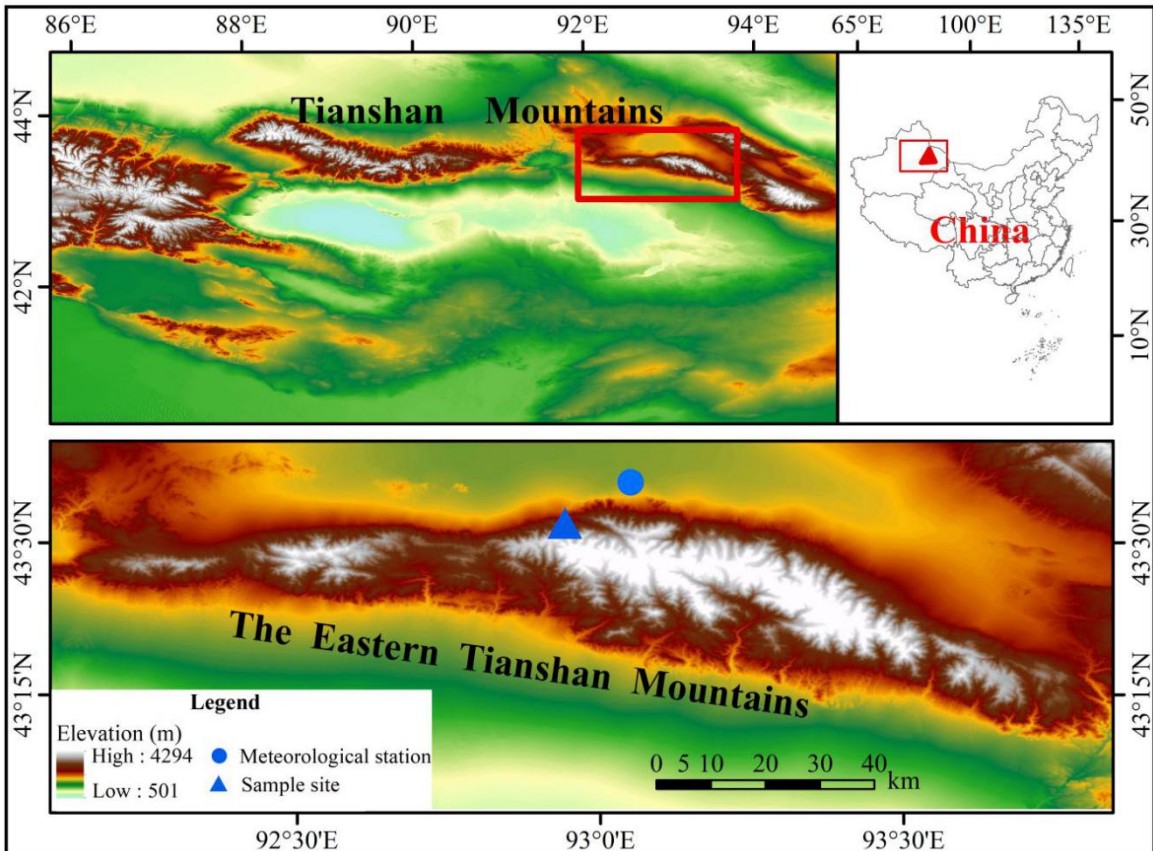

**Figure 1.** Location of the sampling site and the nearest meteorological station (Barkol meteorological station).

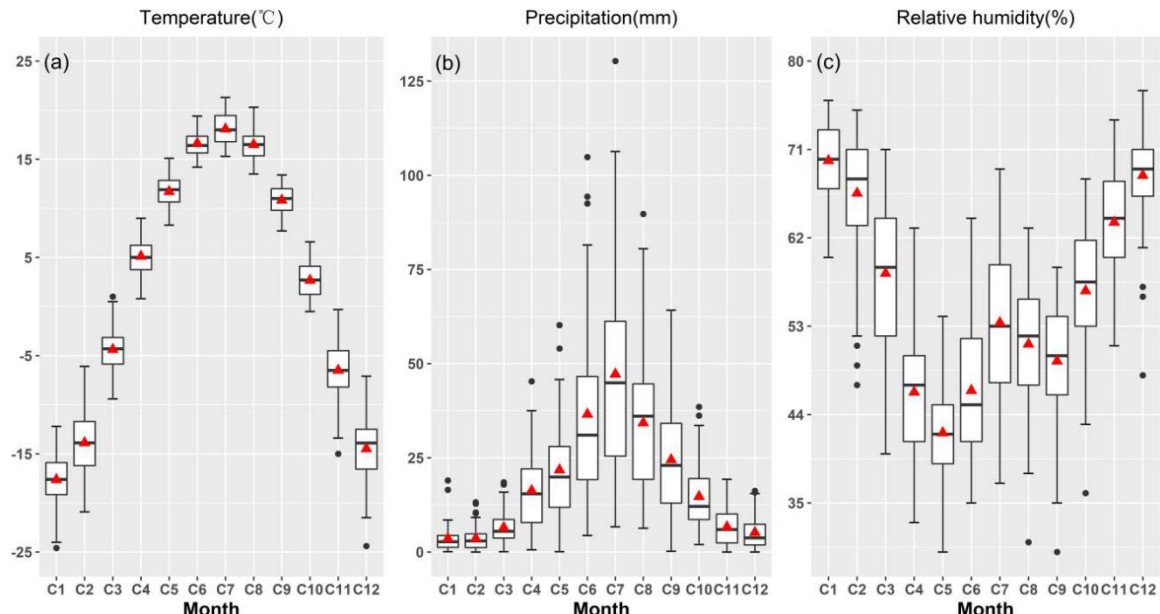

**Figure 2.** Monthly mean temperature (**a**), total precipitation (**b**), and relative humidity (**c**) in Barkol Meteorological Station in eastern Tianshan Mountains from 1958 to 2012. Black dots: outliers; red triangles: average values; lower dash: minimum (excluding outliers); bottom of box: 25% of data; middle line of box: data 50% (Median); top of box: 75% of data; upper dash: maximum (excluding outliers).

### 2.2. Field Sampling

The sampling site was located near the upper forest line on the northeast slope of Barkol in the east of the Tianshan Mountains (43°32′ N, 92°56.5′ E), and the forest was mixed coniferous forests of *Larix sibirica* Ldb. and *Picea schrenkiana* Fisch. & C.A. Mey. This area had an elevation of 2552 m, a slope of 27°, and a forest canopy density of 30%. The average distance between trees was approximately 2.7 m. Healthy Schrenk spruce trees of different breast-height diameters, which were not burnt, damaged, or disturbed by humans, were selected as sample trees. Two cores per tree were collected using an increment borer with a diameter of 5.15 mm at the height of 1.3 m above ground, at an angle of 90 degrees from each other. Thus, a total of 116 tree core samples were collected from 58 living trees in August 2013.

In the laboratory, the core samples were naturally dried, glued into slotted wood boards, and polished with sandpapers of different grit size to allow for the visualization of annual tree rings [44]. The tree-ring width was measured using the LINTAB measurement system (TM6, Rinntech, Heidelberg, Germany) at a resolution of 0.001 mm. Subsequently, the qualities of cross-dated tree-ring width series were determined using the COFECHA program [45]. The age range of the Schrenk spruce cores was from one to 278 years, and we divided all the cores into three age groups (young age group with <100 annual rings, middle age group with 100–200 annual rings, and old age group with >200 annual rings) based on the physiological and age characteristics of Schrenk spruce [46,47]. Finally, the growth trends were removed using a negative exponential curve or linear regression through the ARSTAN program (with a double robust mean) to obtain the standard chronology (*STD*) of the three age groups [20].

To assess the reliability of chronology for the three age groups, eight statistical parameters, including standard deviation (*SD*), mean sensitivity (*MS*), average correlation coefficient (*R*: mean correlation of three series, *R1*: among trees, *R2*: between trees), variance in the first eigenvector percentage (*PC1*), signal-to-noise ratio (*SNR*), and expressed population signal (*EPS*), were calculated for each *STD* (Table 1). Standard deviation (*SD*) estimates the inter-annual variation of each chronology series. Mean sensitivity (*MS*) can measure the richness of the climate information contained in the

chronology and reflect the high-frequency changes of the climate. The correlation coefficient (*R*) measures the synchronization and similarity of width changes between different tree-ring series, and it contains three indicators (*R*: all trees; *R1*: among trees; *R2*: between trees). The R represents the mean correlation coefficient between all sequences in the group; *R1* represents the mean correlation coefficient between two sequences in the same tree; *R2* represents the mean correlation coefficient between trees. In particular, *R2* can indicate common signal presence in growth of particular tree group more clearly. *PC1* shows shared information in the tree-ring samples, and it refers to the amount of interpretation of the variance of the first principal component of the chronology sample within a common interval. Signal-to-noise ratio (*SNR*) is the ratio of climatic information to non-climatic noise in the chronology and can indicate the amount of shared environmental information in the chronology. Lastly, the expressed population signal (*EPS*) reveals the representative of subsample to the entire samples [48–50]. Since these two indicators are important parameters for dendroecological analysis and greatly affected by the sample size, we controlled the sample size of the three age groups (about 20 trees per group) during sampling to reduce the impact of the sample size difference.

**Table 1.** Statistical characteristics of the tree-ring width chronologies of the three age groups of Schrenk spruce (period of dendroclimatic analysis: 1959–2012).

| Statistical Parameters | Young Group | Middle Group | Old Group |
|---|---|---|---|
| Sample depth (core/tree number) | 36/18 | 42/21 | 38/19 |
| Sequence length | 1914–2012 (99) | 1823–2012 (190) | 1735–2012 (278) |
| Mean age | 69 ± 18.34 | 122 ± 18.52 | 236 ± 21.05 |
| Mean tree-ring width | 1.724 | 1.294 | 0.590 |
| Standard deviation (*SD*) | 0.204 | 0.232 | 0.235 |
| Mean sensitivity (*MS*) | 0.249 | 0.256 | 0.261 |
| Correlation coefficient (*R*) | 0.419 | 0.533 | 0.608 |
| Mean correlation among trees (*R1*) | 0.733 | 0.718 | 0.758 |
| Mean correlation between trees (*R2*) | 0.367 | 0.525 | 0.596 |
| The first eigenvector percentage (*PC1*) | 0.496 | 0.559 | 0.639 |
| Signal-to-noise ratio (*SNR*) | 5.776 | 29.668 | 21.734 |
| Expressed population signal (*EPS*) | 0.852 | 0.967 | 0.956 |

*2.3. Meteorology Data*

The meteorological data of daily temperature, precipitation, monthly mean temperature, total precipitation, and relative humidity from 1959–2012 were downloaded from the China Meteorological Data Network (http://data.cma.cn/) of Barkol Meteorological Station in Xinjiang, China. This meteorological station is located on the north slope of the eastern Tianshan Mountains, which is the nearest national alpine meteorological station to the sampling site (distance of about 38 km) with an altitude of 1677 m. Based on the daily temperature and precipitation data, the RClimDex program was used to calculate the extreme precipitations including the annual total precipitation amount with daily precipitation >95% quantile (*R95p*) and the annual total precipitation amount with daily precipitation >99% quantile (*R99p*), which were recommended and developed by the Expert Team on Climate Change Detection, Monitoring, and Indices (*ETCCDI*) in collaboration with the World Meteorological Organization (*WMO*) [51]. Plant drought stress caused by rising temperatures is very significant in the Tianshan Mountains because it is located in the hinterland of the mainland. *SPEI* (standardized precipitation evapotranspiration index) is statistically robust and can indicate the intensity and duration of the drought caused by increased evapotranspiration. Therefore, the *SPEI* is widely used to represent the effects of drought on hydrology and plants under global warming conditions [52]. Because the radial growth of trees was affected not only by the current-year climatic factors, but also by previous-year climatic factors, the climate data from September in the previous year to October in the current year were selected for analyzing the growth–climate relationships [11,53].

*2.4. Data Processing*

2.4.1. Relationship between Radial Growth and Climatic Factors

The inter-annual changes of annual mean temperature, total precipitation, and relative humidity were analyzed by the unary linear regression. The Pearson correlation method was used to calculate the correlation between the three age groups standard chronologies (*STD*) and climate factors (monthly mean temperature, total precipitation, and relative humidity) to determine the main controlling climate factors affecting the radial growth of trees in different age groups. The standardized precipitation evapotranspiration index (*SPEI*) was calculated based on the data of monthly temperature, precipitation, and the latitude and longitude of the Barkol meteorological station [54]. Considering the influence of drought events on the long-term scale, we analyzed the relationships between the standard chronologies (*STD*) of the three age groups for Schrenk spruce and *SPEI* with time scales ranging from one to 24 months using the Pearson correlation method. Moreover, positive *SPEI* values indicate wetness conditions and negative values indicate drought conditions. Therefore, a significant positive correlation between radial growth and *SPEI* shows the impact of drought on trees. Regression, Pearson correlation analysis, and SPEI index calculation were analyzed in R using the packages of "*SPEI*", "ggplot2", and "car".

2.4.2. Pointer Year Selection

In order to analyze the effects of the drought events on the radial growth of the three age groups of Schrenk spruce, the narrow tree-ring years (pointer years) were screened [55]. We used the local standard chronologies of the three age groups to calculate the sliding mean of three years, and we selected the years lower than the mean of the previous three years by 85%. At the same time, the sequences of SPEI values were selected based on the good correlation with the chronologies of the three groups, and we observed changes in the pointer years to ensure that the occurrence of tree pointer years was related to drought.

2.4.3. Resistance Indexes Calculation and Difference Comparison

The resistance indexes of resistance (*RT*), recovery (*RC*), resilience (*RS*), and relative resilience (*RRS*) of the trees in different age groups to interference were calculated according to the following formulae based on the *STD* of tree-ring width in each age group [16]:

$$RT = Dr/PreDr$$
$$RC = PostDr/Dr$$
$$RS = PostDr/PreDr$$
$$RRS = (PostDr - Dr)/PreDr$$

where *Dr* indicates the growth during the dry period (if there are other drought events before and after the comparison of a drought event, it would be quantified as the mean tree-ring width), and *PreDr* and *PostDr* indicate the mean growth during the three years before and after the dry event, respectively. Taking into account the fact that there would be more overlap in drought periods if more than three years after the drought is used as the dividing point (e.g., 1999 and 2003, 2003 and 2008, and 2008 and 2012), three years as a dividing point before and after drought could adequately separate the effects of a single drought event on tree growth [30,56,57]. *RT* represents the ability of trees to resist the drought stress, with a value <1 indicating the reduced growth rate of trees during the drought stress. *RC* represents the ability of the trees under drought stress to restore normal growth after drought, with a value <1 indicating the reduced radial growth of trees after a drought event. *RS* represents the ability of trees to return to their pre-growth status after a drought event, and a high *RS* value indicates a small difference in tree growth between the periods before and after the drought stress. *RRS* is the

weighted resilience to the damage experienced during the drought stress, and a value <0 indicates that the growth of trees is lower after the drought period than during the drought period.

On the basis of the Kolmogorov–Smirnov test, the differences in the resistance indexes (*RT, RC, RS*, and *RRS*) of three age groups were analyzed by one-way analysis of variance (*ANOVA*) and the signifificance level was set to $p < 0.05$. *ANOVA* analysis was performed in R using the "statas" package, and all plots were generated using the "ggplot2" package.

## 3. Results

### 3.1. Statistical Characteristics of Tree-Ring Chronologies in the Three Age Groups of Schrenk Spruce

The statistical characteristics of the tree-ring chronologies of the three age groups showed that the mean tree-ring width index of the young trees was the largest and their *SD* values were the smallest of all age groups, indicating that the young tree grew rapidly with minimal inter-annual variation (Table 1). The *MS* and *R* values of the old and middle-aged groups were significantly higher in the chronology common signal than those of the young group, indicating that the adult Schrenk spruce trees were more sensitive to climate. The highest *EPS* and *SNR* values of the middle-aged group indicated that the chronological samples of the middle-aged trees were the most representative of the population, and the chronology contained more environmental signals. In addition, the *MS* values were greater than 0.2, and the *EPS* values were greater than 0.8, indicating that the three chronologies in different age groups exhibited high quality and sensitivity and were suitable for dendrochronology research.

### 3.2. Inter-Annual Variability of Climate in the Study Regions

The unary linear regression results show the inter-annual change trend of climate factors over the past 54 years (1959–2012). The coefficient of *X* greater than 0 with $p < 0.05$ indicates an increasing trend, and vice versa (Figure 3). The inter-annual variabilities of mean temperature, total annual precipitation, and relative humidity were significant, showing the increasing trends of the mean temperature with 0.617 °C/decade and the annual total precipitation with 10.2 mm/decade, and the decreasing trend of the relative humidity with −1.56 mm/decade from 1959–2012. In addition, there was a very significant negative correlation between the relative humidity and the mean temperature ($R = −0.7906$, $p < 0.01$), and there was not significant correlation between the relative humidity and the total precipitation, indicating that the main reason for the dry climate in the study regions was temperature increase (Table 2). Moreover, the annual total precipitation and extreme precipitation also showed a significant positive correlation (R95p: 0.6730, R99p: 0.5156, $p < 0.01$), indicating that the increase in total precipitation was accompanied by an increase in extreme precipitation.

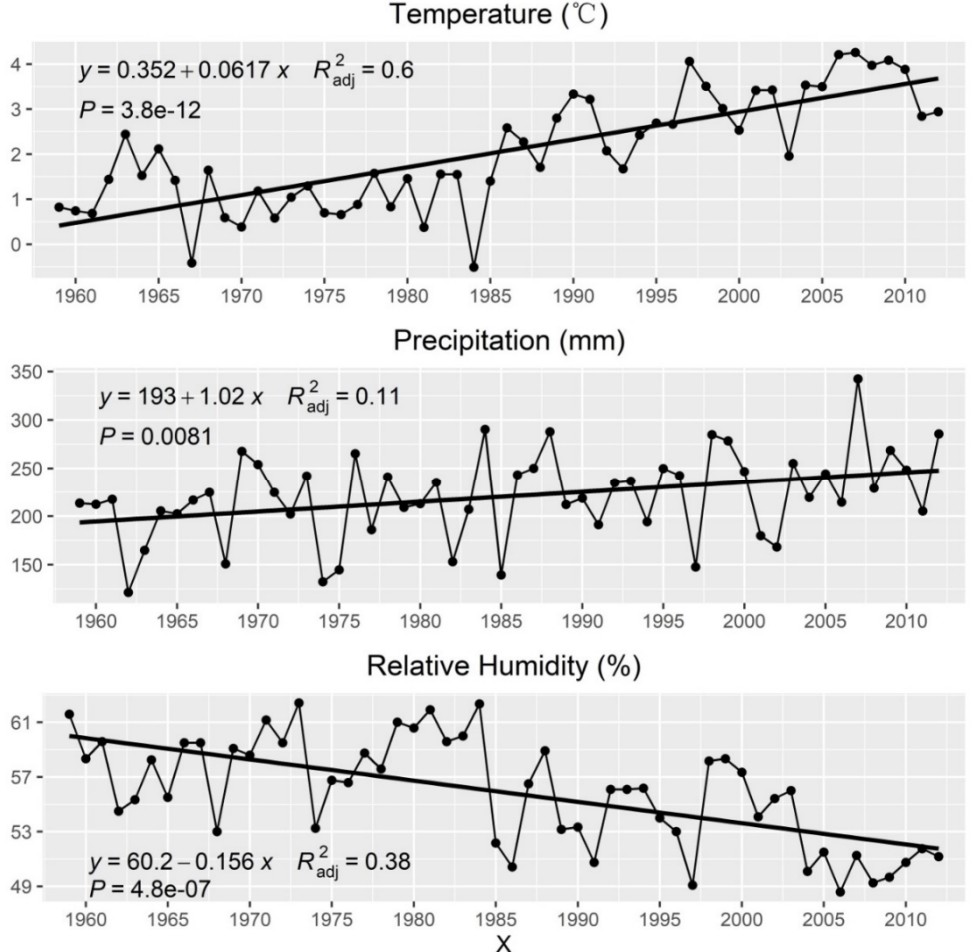

**Figure 3.** Inter-annual variation trends of mean temperature, annual total precipitation, and relative humidity from 1959–2012 using the records from the Barkol meteorological station. The thick solid lines represent the simulated variation trends by linear regression. P: the significance level of the $X$ coefficient; $R^2_{adj}$: the adjusted coefficient of determination.

**Table 2.** Correlation coefficients between temperature, precipitation, R95p, R99p, and relative humidity.

|  | Temperature | Precipitation | *R95p* | *R99p* | Relative Humidity |
|---|---|---|---|---|---|
| Temperature | 1 | 0.1235 | 0.3299 * | 0.2845 * | −0.7906 ** |
| Precipitation |  | 1 | 0.6730 ** | 0.5156 ** | 0.0956 |
| *R95p* |  |  | 1 | 0.6456 ** | −0.1895 |
| *R99p* |  |  |  | 1 | −0.2136 |
| Relative humidity |  |  |  |  | 1 |

* Significant at the 0.05 level. ** Significant at the 0.01 level.

*3.3. Relationship between Radial Growth and Climatic Factors in Three Age Groups*

3.3.1. Relationship between Radial Growth and Temperature in Three Age Groups

The radial growth of trees in three age groups was negatively correlated with temperature, but with different $R$ values in every month (Figure 4). For example, significant negative correlations were observed between the chronology of young trees and temperature in September ($R = -0.323$, $p < 0.05$) and November ($R = -0.349$, $p < 0.01$) of the previous year and in March ($R = -0.289$, $p < 0.05$), May ($R = -0.282$, $p < 0.05$), June ($R = -0.415$, $p < 0.01$), August ($R = -0.293$, $p < 0.05$), and September ($R = -0.277$, $p < 0.05$) of the current year. Furthermore, the chronology of the middle-aged

trees and mean temperature showed a significant negative correlation from September to November of the previous year and from February to October of the current year ($R = -0.282$ to $-0.610$, $p < 0.05$). In addition, the chronology of the old trees and mean temperature showed a negative correlation in May of the current year ($R = -0.266$, $p < 0.05$).

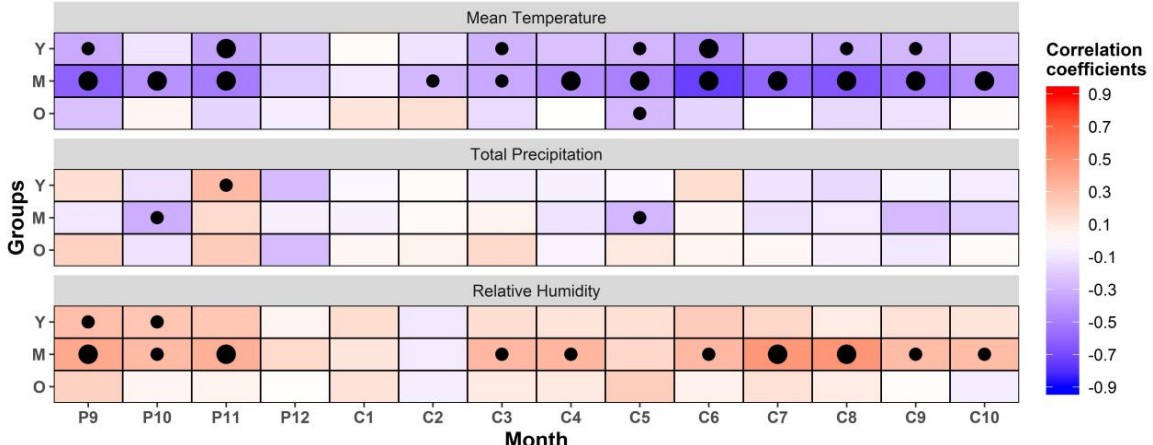

**Figure 4.** Correlation coefficient between the radial growth and mean temperature, total precipitation, and relative humidity in the Schrenk spruce trees of three age groups. Y: young group; M: middle-aged group; O: old group; red: positive correlation; blue: negative correlation; large black dots: significance level <0.01; small black dots: significance level <0.05; P: previous year, C: current year (for example, P9: September of the previous year, C1: January of the current year).

### 3.3.2. Relationship between Radial Growth and Moisture Conditions (Precipitation and Relative Humidity) in Three Age Groups

It can be seen that the relationships between the *STD* of the three age groups of Schrenk spruce trees and moisture conditions (total precipitation and relative humidity) were different (Figure 4). The chronology of the old trees was not significantly correlated with total precipitation and relative humidity, whereas the chronology of the young trees was significantly positively correlated only with total precipitation in November ($R = 0.321$, $p < 0.05$) of the previous year. However, the chronology of the middle-aged trees was significantly negatively correlated with total precipitation in October ($R = -0.313$, $p < 0.05$) of the previous year and May ($R = -0.218$, $p < 0.05$) of the current year.

The correlations of the radial growth of the young and middle-aged trees with relative humidity were different from their correlations with precipitation. The radial growth of the young trees showed a significant positive correlation with relative humidity in September ($R = 0.299$, $p < 0.05$) of the previous year and October ($R = 0.266$, $p < 0.05$) of the current year. The radial growth of the middle-aged trees and relative humidity had a significant positive correlation from September to November ($R = 0.319$–$0.403$, $p < 0.05$) of the previous year and in March ($R = 0.334$, $p < 0.05$), April ($R = 0.338$, $p < 0.05$), and June to October ($R = 0.31$–$0.486$, $p < 0.05$) of the current year.

### 3.3.3. Relationship between Radial Growth and SPEI in Three Age Groups

The results of the Pearson correlation between the *STD* of the three age groups and *SPEI* in one- to 24-month scales were different (Figure 5). The young (53.12%) and old (22.92%) trees had the highest positive correlation frequency with *SPEI* when compared to the middle-aged trees (non-significant correlation). In addition, the young and old trees were affected by drought in various scales from November of the previous year to December of the current year. These results indicated that the young and old trees were more sensitive to drought stress than the middle-aged trees.

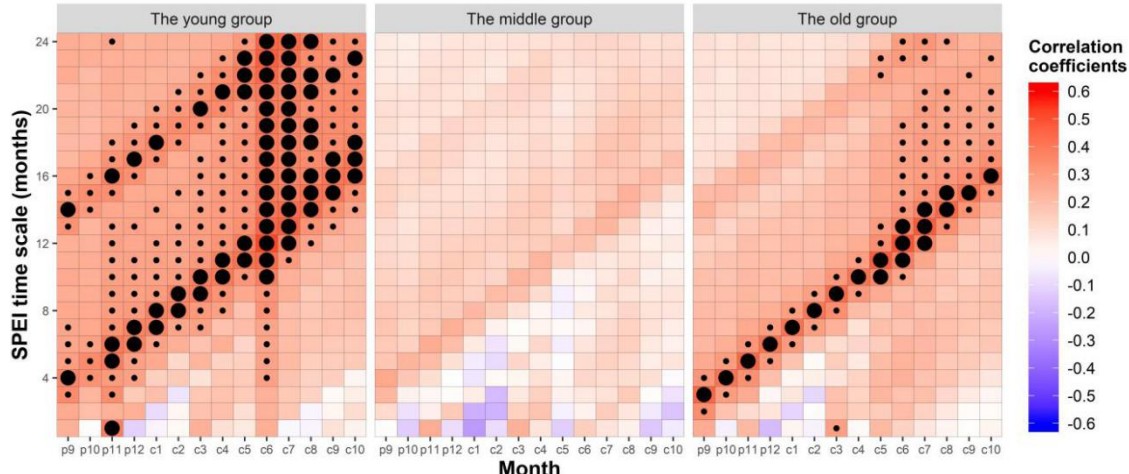

**Figure 5.** Correlation coefficient between radial growth and standardized precipitation evapotranspiration index (*SPEI*) in the three age groups of Schrenk spruce. The *X*-axis is the month, and the *Y*-axis is the duration of the drought. Red: positive correlation; blue: negative correlation; big black point: significance level <0.01; small black point: significance level <0.05; P: previous year; C: current year (for example, P9: September of the previous year, C1: January of the year, a big black spot with *X* = C9 and *Y* = 4 indicates that the standard chronology of the young trees has a significant positive correlation with the SPEI lasting for four months in the previous September).

### 3.4. Resistance Variation of Radial Growth to Drought in Three Age Groups

The *STD* of the three age groups had a relatively consistent inter-annual variability trends, and the correlation coefficient between the two groups reached significance (R young–middle: 0.746; R young–old: 0.6239; R middle–old: 0.4873; *p* < 0.01). The narrow tree-ring years of the three age groups were screened based on the STD variability of the tree-ring width abnormally 15% below the average of the previous three years (young trees: 1967, 1974, 1978, 1986, 1998, 1999, 2003, 2008, and 2012; middle-aged trees: 1967, 1974, 1978, 1981, 1986, 1998, 1999, 2002, 2003, 2008, and 2012; old trees: 1967, 1972, 1974, 1978, 1980, 1986, 1998, 2001, 2003, 2008, and 2012). Since the recovery of trees in 2012 cannot be calculated due to the tree-ring width being up to 2012, eight public years were selected (1967, 1974, 1978, 1986, 1998, 1999, 2003, and 2008). Then, they were compared with the SPEI in June with a 12-month scale, using accumulated and the standard chronologies of three age groups, only when the *STD* decreased by more than 15% and the *SPEI* corresponding to a low value could ensure that the extremely narrow year. In the absence of low *SPEI* values, *STD* decreased by more than 15% (such as 1963 and 1991), which might have been caused by other environmental disturbances (pest outbreaks, fires, geological disasters, etc.). Finally, the calculation of resilience required a three-year average of the tree ring width after a dry year, and seven pointer years (1967, 1974, 1978, 1986, 1998, 2003, and 2008) were determined as drought years (Figure 6).

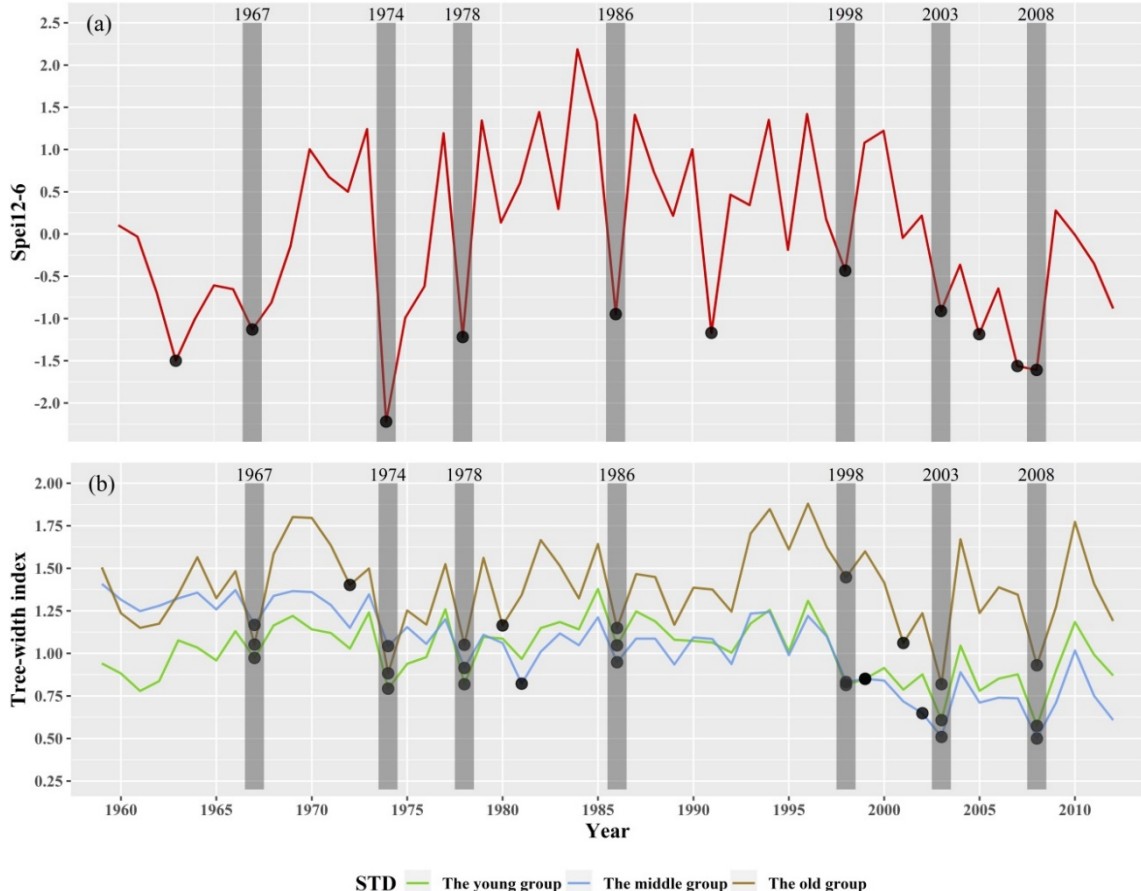

**Figure 6.** Inter-annual variability of the standardized precipitation evapotranspiration index (*SPEI*) values in June with a 12-month scale (**a**) and tree-ring width chronologies of the three Schrenk spruce age groups (**b**). The black dots indicate the selected pointer years; the rectangular shadings indicate the common pointer years of the SPEI and chronologies.

Resistance indexes (*RT*, *RC*, *RS*, and *RRS*) were calculated to determine the resistance of Schrenk spruce to drought (Figure 7). By comparing the resistance (*RT*) values of the three age groups through one-way *ANOVA*, there were significant differences between the four resistance indicators of the middle-aged group and the old-aged group ($p < 0.05$), and the young-aged group was not significantly different from the middle-aged and old-aged groups (Table 3). In general, the resistance of the middle-aged group was higher (0.841) than that of the young (0.804) and old (0.775) groups. Contrary to the *RT* values of the three age groups, the *RC*, *RS*, and *RRS* values of the old (*RC*: 1.316, *RS*: 1.005, *RRS*: 0.23) and young (*RC*: 1.263, *RS*: 1.002, *RRS*: 0.198) groups were the higher than those of the middle-aged group (*RC*: 1.149, *RS*: 0.949, *RRS*: 0.108).

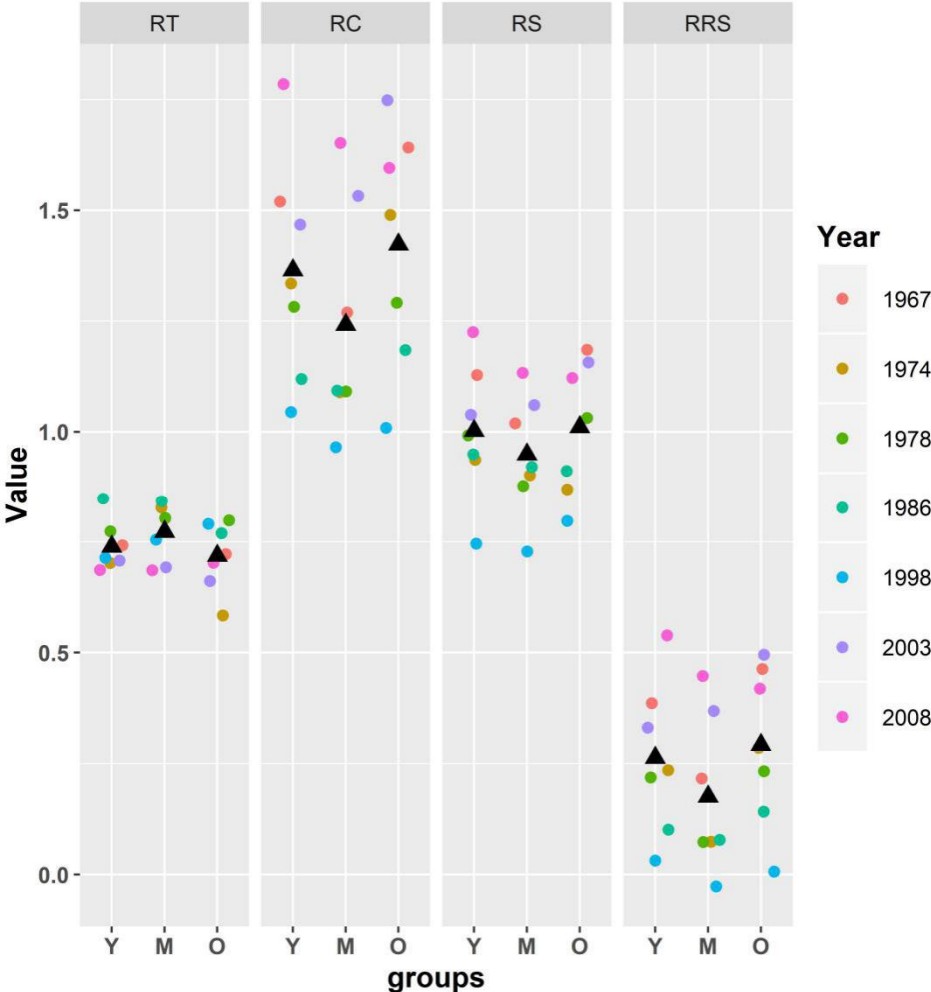

**Figure 7.** Scatter plots of the resistance indexes (*RT*, *RC*, *RS*, and *RRS*) of the Schrenk spruce trees of the three age groups in seven drought years. The black triangles indicate the average of values obtained in seven different years. *RT*: resistance; *RC*: recovery; *RS*: resilience; *RRS*: relative resilience; Y: young group; M: middle-aged group; O: old group.

**Table 3.** Discrepancy analysis of the resistance indexes of Schrenk spruce trees in three different age groups by one-way analysis of variance (*ANOVA*).

|   | Resistance (*RT*) | Recovery (*RC*) | Resilience (*RS*) | Relative Resilience (*RRS*) |
|---|---|---|---|---|
| *Y* | 0.804 ± 0.0167 [ab] | 1.263 ± 0.0461 [ab] | 1.002 ± 0.0272 [ab] | 0.198 ± 0.0336 [ab] |
| *M* | 0.841 ± 0.0180 [a] | 1.149 ± 0.0462 [b] | 0.949 ± 0.0239 [b] | 0.109 ± 0.0345 [b] |
| *O* | 0.775 ± 0.0179 [b] | 1.316 ± 0.0490 [a] | 1.005 ± 0.0290 [a] | 0.230 ± 0.0346 [a] |

Different letters in the same column indicate the significant difference ($p < 0.05$). Y: young group; M: middle-aged group; O: old group.

## 4. Discussion

### 4.1. Response Discrepancy of Radial Growth to Drought in Three Age Groups of Schrenk Spruce

The radial growth fluctuations of trees are largely caused by climatic changes. In the present study, we analyzed the relationships between the standard chronologies of Schrenk spruce and environmental variables, including mean temperature, total precipitation, and relative humidity (Figure 4). The chronologies of three age groups had a consistent negative correlation with temperature,

but a positive correlation with relative humidity, indicating that the high temperature-induced drought stress was the main limiting factor for the radial growth of trees in the three age groups. This response pattern was also found in other coniferous tree species in the same study region [25,54].

A rapid increase in temperature and lack of available water were the main reasons for limiting the radial growth of trees in the eastern Tianshan Mountains. Despite the warm-dry to warm-wet transition in the Tianshan Mountains [58,59], the increase in temperature was significantly higher than that of precipitation, and the increase in total precipitation was accompanied by a form of extreme precipitation in the eastern Tianshan Mountains (Figure 3 and Table 2). The increase in the intensity of short-term precipitation would increase the probability of soil erosion and reduce the soil infiltration, absorption, and transformation of precipitation, which would indirectly lead to a reduction in the water available to the plant's roots [60]. In the growing season, the temperature rises and the lack of available water will lead to physiological dehydration and physiological drought in plants [61]. This greatly increases the survival pressure of trees in sensitive and vulnerable arid and semi-arid areas [62,63]. Drought stress will directly affect the transpiration and respiration of trees, leading to a reduction in the rate of photosynthesis, which will affect the carbon acquisition and carbon allocation of trees, and eventually narrow the width of tree rings [64]. For gymnospermns [23,28,65]. An increasing number of reports are showing declines in tree growth and increased mortality ca plants, under repeated drought stress, the carbon balance and hydraulic performance of the plant will continue to deteriorate, which will eventually cause irreversible damage to plant tissues and orgaused by drought in different forest types and regions of the world [66,67].

The impact degree of drought stress on individual trees is not only related to forest density and stand composition, but also closely depends on tree age [22,56]. Therefore, a drought caused by increasing temperatures has a strong growth restriction effect in trees of different age groups; in particular, the young and old groups had more significant positive correlation with the drought indicator SPEI in our study regions (Figure 5). Young trees suffered more severe water stress and greater mortality risk during the dry periods, suggesting that they were more sensitive to climate in severe environmental conditions because of the loss of their internal resource storage resulting from their "overspending" strategies in photosynthesis and transpiration and their limited ability to access resources with underdeveloped lateral roots [68,69]. Meanwhile, the older trees had complex water channels and showed more water consumption with increasing tree height [29]. Even if the height growth was maximized at an early stage of development, the older trees were still subject to persistent drought stress because of the increase in their water transpiration with continuous diameter growth and crown expansion [70].

## 4.2. Resistance Discrepancy of Radial Growth to Drought in Three Age Groups of Schrenk Spruce

The correlation analysis results of radial growth in different age groups of Schrenk spruce and climate factors were contradictory, showing that the middle-aged trees were sensitive to hydrothermal climatic conditions (mean temperature, total precipitation, and relative humidity), but had no significant correlation with the comprehensive drought index SPEI (Figures 4 and 5). However, the correlations of old and young trees were only significant with SPEI, but not with hydrothermal climatic factors. These opposite results might have been produced by the differences in the drought resistance of different age trees.

The effect of age on the resistance differences of Schrenk spruce trees in different age groups was determined by analyzing the dynamic characteristics of a Schrenk spruce population [71]. The average resistance of Schrenk spruce trees under drought events was in the order middle age > young age > old age (Figure 7 and Table 3). The highest resistance to the onset of drought stress partially neutralizes the effects of drought on radial growth, thus showing no significant correlation between radial growth of the middle-aged group and SPEI. Meanwhile, the middle-aged trees had three advantages in resisting drought stress. Firstly, the Schrenk spruce trees of different age groups need different resource allocation strategies to improve their own adaptability in the community, and the middle-aged Schrenk spruce

have strong drought tolerance with abundant resources stored in stems and roots because of their strong physiological activity and cell vitality [72,73]. Secondly, the middle-aged trees showed stronger water absorption and storage capacity than the young trees, because water in the deeper soil could be absorbed efficiently with a well-developed root system as in the middle-aged trees [43,67]. Finally, the middle-aged trees were more competitive than the old trees because the older tree population exhibited phenomena of decreased over-ripe resistance, reduced physiological activity, and increased root lignification with increasing age [72,74]. Therefore, the older and younger trees showed lower resistance to drought stress than the middle-aged trees because of their reduced efficiency of acquisition and utilization of nutrient and water resources [75].

The *RC*, *RS*, and *RRS* indexes could demonstrate the ability of plants to restore physiological functions from drought stress in a different manner. However, those three indexes in different age groups in terms of response to drought showed opposite results to resistance index, with the order old trees > young trees > middle-aged trees (Figure 6 and Table 2). This might be related to the intrinsic physiological characteristics of the different age trees of Schrenk spruce. On one hand, the slower growth rate and "conservative" nutrient utilization strategy of older trees would guarantee more resources for subsequent recovery after a drought event [29]. On the other hand, the old trees developed a complete drought response strategy as a result of the pronounced "cumulative effect" of several drought events on their radial growth [57,72]. Compared to middle-aged trees, the young trees with smaller total biomass could recover quickly after drought due to their lower requirement of resources like carbon, nutrients, and water [76,77]. In addition, the lower recovery capacity of the middle-aged trees after experiencing drought stress was related to consuming more resources against the drought stress at the beginning of the drought, showing stronger resistance and weaker resilience [73]. Therefore, trees in different age groups adopted different ecological response strategies to drought, suggesting that the middle-aged trees paid more attention to resistance during a drought event, while the young and old trees paid more attention to recovery after drought.

## 5. Conclusions

Associated with an increase in global warming, drought stress is a main factor limiting forest ecosystems in Central Asia. The radial growth of Schrenk spruce trees of different age groups in the eastern Tianshan Mountains is mostly controlled by high temperature-induced drought events, as shown by the analysis of their growth–climate relationships and resistance indexes. However, the trees of different age groups had different adaptation strategies to cope with drought stress. The radial growth of the middle-aged trees was less correlated with drought indicators, and they exhibited higher resistance and lower resilience to drought, but the young and old trees were more affected by drought, exhibiting lower resistance and higher resilience after drought. With an increase in the frequency and intensity of drought in the context of global warming, the structural composition and dynamics of forest ecosystems in Central Asia might be significantly impacted in the future based on the age-dependent discrepancies of climate responses and drought tolerances of trees. Therefore, the age effect of trees should be considered when studying the relationships between tree radial growth and climate factors, and the adaptation diversity of different age trees should also be taken into account when formulating the scientific forest protection measures under future climate change scenarios.

**Author Contributions:** Study conception and experimental design, L.J. and X.L.; acquisition and processing of samples, L.J., X.L., S.W., and K.C.; analysis and interpretation of data, L.J. and X.L.; drafting of manuscript, L.J. and X.L. All authors have read and agreed to the published version of the manuscript.

**Funding:** This research was funded by the National Natural Science Foundation of China (41861006 and 41630750), the Research Ability Promotion Program for Young Teachers of Northwest Normal University (NWNU-LKQN2019-4), and the Scientific Research Program of Higher Education Institutions of Gansu Province (2018C-02).

**Acknowledgments:** We also thank the anonymous referees for helpful comments on the manuscript.

**Conflicts of Interest:** The authors declare no conflicts of interest.

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
