# Peer review of "Radial Growth Adaptability to Drought in Different Age Groups of Picea schrenkiana Fisch. & C.A. Mey in the Tianshan Mountains of Northwestern China"

_forests, doi:10.3390/f11040455_

Round 1
Reviewer 1 Report
My comments are included in the attached file below.

Author Response
My responses are included in the attached file below.

Reviewer 2 Report
During revision of the manuscript, its authors responded to the most of comments adequately. There are several questions still:
1. All three age groups' chronology seem to have similarity in their variation, as is evident from Fig. 6 (bottom panel). It would be useful to show it quantitatively as correlations between them over common period and over period of dendroclimatic analysis.
2. In the Fig. 6, mean values of chronologies seems to be different from 1, as it should be for standard chronologies. Check if this plot is correct. Perhaps, you should show full length of chronologies, probably in appendix.
3. Also, after closer comparison between climatic dynamics in Fig. 3 and chronologies in Fig. 6, I think that higher/lower dendroclimatic correlations between age groups can be caused not only due to actual variation of sensitivity, but also due to common or opposite directions of long-term trends in compared time series. You can try and check it if you delete linear trends from both, and calculate correlations between detrended series.
4. In the Fig. 6, please mark with dots all years when SPEI is extremely low, and all pointer years in chronologies (drop of tree-ring width), then state in the caption that you analysed years when both drought and growth depression in two or three age groups were observed, except 2012. It will make figure more understandable.
5. EPS and SNR both depend on number of samples, you should show in text clearly that you take it into account when compare chronologies. Inter-tree correlation indicates common signal presence in growth of particular tree group more clearly; EPS and SNR characteristics are important in regards to suitability of chronology for dendroecological analysis.
Author Response
My responses are included in the attached file below

Reviewer 3 Report
I believe the authors have revised the manuscript as needed and I feel the changes are adequate for publication. Good work!
Author Response
Thank you for your recognition of our research work. And we have further modified the language so that readers can understand it better.
This manuscript is a resubmission of an earlier submission. The following is a list of the peer review reports and author responses from that submission.
Round 1
Reviewer 1 Report
Please see the attachment.

Reviewer 2 Report
The authors present a study assessing tree-ring growth response of Schrenk spruce across three age classes in the Tianshan Mountains of China. They found that drought stress was the main driver of radial growth in this species and region. Moreover, this effect was most pronounced in the young and old aged trees, having less radial growth under drought than middle aged trees. The authors also examined metrics of resistance, recovery, resilience, and relative resilience in tree-ring growth and found differing effects by age class. The results could inform management of Schrenk spruce in this region.
Overall, I found the topic compelling but the manuscript, as written, did not seem to deliver the necessary justification and information. I found no specific hypotheses regarding tree age, radial growth, and drought. Moreover, the justification and methods were drastically lacking in detail. Thus, it was challenging to understand the logic, interpret the results, and ultimately provide relevant feedback on the study. I provide some broad comments and more specific comments below in hopes they can help improve a future iteration of the manuscript.
Broad/major comments
Suggest reviewing and likely including the reference below:
Cailleret, M., S. Jansen, E. M. R. Robert, L. Desoto, T. Aakala, J. A. Antos, B. Beikircher, C. Bigler, H. Bugmann, M. Caccianiga, and V. Cada. (2017). A synthesis of radial growth patterns preceding tree mortality. Global Change Biology 23(4), 1675-1690.
Introduction:
Would like some hypotheses here about tree age, radial growth, and drought response. As written, it is unclear why age matters in tree-ring growth or why there might be conflicting patterns of tree-ring growth among different age classes.
Materials and Methods:
More detail is needed here on the sampling site and more clarity on the choice of tree-ring methods. There is virtually no information on the forest type. What is the tree density, basal area, other common species within the sampling location? What age do Schrenk spruce usually reach? Was there any management in the study area? Also, what was the sampling design for collecting the 58 tree cores (e.g., sampling area, plot dimensions/area, number of plots, layout of plots, etc.)? Density of trees can influence radial growth patterns thus, it is critical to have a better understanding of the potential competitive effects of the sampled trees vs. just interpreting as climatic effects. Did the trees in different age groups occur in different conditions or were they all sampled randomly from the same area and conditions?
How was ‘extreme drought’ defined and differentiated from non-extreme drought? The methods for identifying drought events within the tree ring chronologies need more detail and clarification. It seems bold to discuss ‘extreme’ drought without a clear distinction from less extreme drought or dry/hot periods. It would be helpful to see temperature, precipitation, and SPEI data temporally (e.g., rather than aggregated by month) to see magnitude of specific drought events in the region.
Regarding age effects, were tree rings from a certain fixed time period compared (e.g., most recent 100 years) rather than all rings for each age group? For example, the middle age group would have tree rings pertaining to its ‘young’ age and ‘middle’ age…and the old age group would have tree rings pertaining to its ‘middle’ and ‘young’ ages in addition to its ‘old’ age. Thus, comparing all rings from each age group might produce confounding results.
Discussion
A number of claims are made here and I am not certain they are all supported by the data or specifically how they relate to the results. Moreover, since there were no clearly stated hypotheses, it is difficult to assess whether the authors achieved the goals they intended to and if the discussion points accurately relate to the results presented.
Minor comments:
Lines 91-92: What studies are you referring to here? This statement is confusing and needs a more detailed description.
Lines 117-118: Why were cores oriented 120 degrees from each other (i.e., as opposed to 180 degrees or 90 degrees)?
Line 120: Please clarify what is meant by a ‘fixed wooden tank’.
Lines 130-136: Provide more detail on why these specific eleven metrics were used and what relevant information each measure provides.
Lines 141-143: Please describe how relative humidity data was summarized for analyses.
Lines 184-185: How can young trees have a fast average growth rate yet have small annual ring width? This seems contradictory.
Lines 238-243: I had a very hard time understanding what this paragraph was trying to convey about the data. I don’t fully understand why the seven years listed are ‘extreme’ drought years as opposed to just drought years.
Lines 291-299: This is the first location I have seen descriptions of why tree age matters in radial growth response. This should be covered much more thoroughly in the Introduction and Methods and presumably used to inform hypotheses.
Table 1: Suggest showing standard deviation of tree age (i.e., with mean age).
Figure 1: The labels on the map could be placed in locations that are easier to read or could be removed and instead mentioned in the caption. Also, I assume BLK refers to the meteorological station. Make this clearer in the figure. I think this figure might be more useful if the portion with the colored elevation gradient was zoomed in closer to the sampling location rather than generally showing the Tianshan Mountains. The Tianshan Mountain range could be labeled generally in the smaller inset figure instead. I recommend using a different color for the sampling plot triangle marker. The red is difficult to discern among the mid-elevation color around it on the map.
Figure 2: Mention where the data were collected here (i.e., the meteorological station) and what the red triangles reference (i.e., mean monthly precipitation?). Also, why boxplots for the temperature but not for the precipitation?
Figure 3: Vapor pressure deficit (VPD) may be a better metric than relative humidity. Many tree physiology studies have shown strong correlations between VPD and transpiration.
Figure 4: This figure is challenging to interpret given it has so many temporal drought index levels. The main thing I gathered from all this information is that annual growth in the middle age group does not correlate to any drought index value. Your prior knowledge and hypotheses should allow you to select those SPEI drought levels that are most relevant rather than looking at all levels from 1-24 months…or, present only those drought levels which directly support your hypotheses.
Reviewer 3 Report
The statistical presentation is excellent. However, the fundamental question being asked should not have been answered with tree ring width -- it should have been answered with basal area increment growth which would allow a direct comparison across tree age classes. Yes, the ring widths are 'smaller' in old trees, but the production/growth is measured by the increase in that small tree ring width over the whole circumference of the tree, as is the 'large' tree ring width of the younger tree is integrated over a relatively small diameter...
The initial presentation is nicely written, but the discussion refers to a number of other papers published on the overall study, which should have been described in the methods as background info. The discussion also cites a number of other studies as supportive of the results of this paper, yet other species in other ecosystems are used that may or may not be relevant to the point being made. I have noted this in the paper.
There is no acknowledgement that trees adapt to stress (or they die). The analysis presents the drought indices, I think although it was not explicitly stated in the methods, that were averaged over the last 50 years. The drought experienced in 2003 was the lowest in the record presented. the tree ring index (index? width?) was the same in 2009 as in 2003 for young and middle aged trees suggested that either a threshold response was reached for the xylem structure of both young (100 yrs!) and middle age trees (100-200 yrs!), or that both age classes adapted to that level of stress. to me, fig 5 shows that the older trees were the most resilient, hands down, than the other 2 age classes. The age classes are too broad. Conifers can have a 'youthful' physiology up to about 40 yrs old. About >~75 yr they have the physiology of 'mature' trees : acquisition, allocation, resilience to stressors. Old trees: > ~180 yrs. We don't have enough information on the age structure of the tree groups, the soils, the proximity of the meteorological station to the site, the sampling 'points' -- how close were they to the station? (etc) to be able to interpret this data set and analysis as presented.
There are many points made in the text---

Reviewer 4 Report
The manuscript "Radial growth adaptability to extreme drought in different age groups of Picea schrenkiana in the Tianshan Mountains of northwestern China" by Liang Jiaoa, Xiaoping Liu, Shengjie Wang and Ke Chen describes research of climatic response and growth stability of three age classes of Schrenk spruce trees in the eastern Tianshan Mountains, China. Authors compared standardized chronologies of tree-ring width (TRW) with mean temperature, precipitation, and relative humidity from the closest meteorological station and with SPEI drought index, and then analyzed age dependence of growth stability coefficients (resistance, recovery, resilience, and relative resilience). The research is novel and methodological approach is reasonable. Findings and conclusions are of interests in regards to tree growth–climate interactions and also for forest management.
General comments.
There are several issues and questions that should be addressed before publication of the manuscript. First of all, English language quality (grammar and style) can be improved. Some suggestions in this regard are in the minor comments below, but I am not a native English speaker.
In regards to meteorological data, distance between meteostation and sampling site is not stated; this distance, presence of mountain ridges/peaks between station and sampling site, and difference in elevation by ~900 m may decrease relevance of the meteostation series in regards to this particular sampling site. You should discuss this issue and measures taken to solve it or take into account. Temperature lapse rate direct observations in the study area or general estimation -0.65°C per 100 m of elevation can be used to correct temperature series and see, e.g., length of the vegetation season and summer maximum of the temperature at the sampling site. On the other hand, distance and landscape obstacles may be mentioned in discussion as one of the reasons decreasing spruce response to precipitation (since its field is more stochastic than temperature) and relative humidity. More thorough investigation of local climate will be useful also to consider if there can be direct negative impact of temperature on tree growth in the summer, i.e., heat stress, as additional reason for response to temperature to be stronger than reactions to precipitation and relative humidity. Also, you may show more detailed map of the study area (on local scale) in the Fig. 1, where distance and landscape between sampling site and meteostation will be shown clearly.
I am confused about autocorrelation coefficient. ARSTAN program output is the first-order autocorrelation (AR-1, i.e., correlation of chronology with itself shifted by one year), did you use this characteristic? Then, if STD chronologies were used, why AR is close to zero? Usually in STD chronologies it has positive values > 0.5; small and especially negative autocorrelation is more typical for residual (RES) chronologies developed by suppression of autocorrelation. For the purposes of your research, I recommend usage of STD or RES chronologies in dendroclimatic analysis, and STD or average measured (raw) TRW for growth stability analysis. For calculation of Rt, Rc, Rs, and RRc coefficients, most of studies use raw TRW or BAI series (that have less age trends), but comparison between patterns obtained with usage of STD and raw chronologies can be interesting here or in further research.
In regard to usage of 3-year interval before drought and post-drought, since there are less than 6 years between some droughts, these intervals are overlapped. I understand that lesser periods are unreasonable, but you should at least mention this fact and its possible impact on your findings. In dendroclimatic analysis, did you consider investigation of seasonal climatic series (temperature, precipitation, relative humidity), e.g., for previous September-November and current March-October? Their correlations with STD may be stronger than monthly series. In this case, seasons can be defined as in Williams et al. (2013; see methodological description in Supplementary material). In analysis of growth stability, you may add investigation of its dependence on drought severity (SPEI 12.6 or other climatic series).
Minor comments.
L5. Please add affiliation of Shengjie Wang.
L38. What is a meaning of “water crisis” here? Did you mean “water stress”?
L42. I suggest writing word “annual” in parentheses, because it is a clarification of temporal resolution, like this: “high (annual) resolution”.
L55. I suggest replacing “original” with “undisturbed” or fully omitting it.
Fig.1. What does label “BLK” mean?
Fig.2. In the figure caption, swap “precipitation” and “mean temperature”, since they are messed up. I recommend addition of more information: 1) make clear in the caption this series are from Barkol meteostation; 2) perhaps, you can add relative humidity as a new panel (c); 3) describe all elements of plots as a legend or in the caption (i.e., if boxes show mean or median as horizontal line, standard deviation or standard error as box, full range of XX percentiles as lines, used definition of outliers showed as points; in the right panel, define meaning of points and triangle markers).
L117. “Increment borer” is the more often used term. Also, it is not clear why 120° angle was used.
L123. You mark trees with >200 rings as old. What is lifespan of the species in general and in the study region?
L134. EPS shows common signal in general, which can contain not only climatic influence, but also impact of other factors and events common for all or most of trees (like forest stand dynamics, pollution, disturbances caused by fires, pests, etc.). Please rephrase the sentence.
Table 1. Please replace “common period” with “period of dendroclimatic analysis”, since for your chronologies, common period is 1914-2012.
L144. Please state source of SPEI series: grid data base like http://spei.csic.es or https://climexp.knmi.nl and respective geographic coordinates, or calculation from meteostation data and how you did it (with proper citations and used program). In the first case, you should add geographic grid cell (rectangle) or its center (marker) on the map in Fig.1.
L157-161. Please combine from L157-161 and L238-241more detailed description of this procedure; in results (L238-241) state only that you selected 12-month SPEI in June as climatic variable for pointer years’ selection. Also, state explicitly that you used here local STD chronology (not individual tree series), and correct term “pointer years” throughout the manuscript. Also I suggest to cite reference with used definition of pointer year, like work of Schweingruber et al. (Schweingruber, F.H., Eckstein, D., Serre-Bachet, F., Braker, O.U., 1990. Identification, presentation and interpretation of event years and pointer years in dendrochronology. Dendrochronologia 8, 9‒38).
L163. Please cite source of equations, like Lloret et al. (2011).
L189. SNR and EPS depend on sample size, they have higher values for mid-age group since it is represented by more trees. Common signal is higher in old trees (values of inter-series correlation coefficients). AC characteristic is not about climatic signal at all, vice versa, high autocorrelation decreases climatic signal; also, see major comment about my concerns on its values.
L212-213. Why you mention here different study regions?
L232. The term “vulnerable” is more associated with the risk of tree mortality; I suggest instead use term “sensitive” as indicator of the more pronounced immediate reaction.
Fig. 4. Clearly state, if month marked on the horizontal axis is the last month for averaging of SPEI (e.g., that cell marked with 6 months time scale and c10 on the month axis is SPEI from May to October of current year) as it seems to be. It is not clear from the first glance for readers that don’t use SPEI themselves.
Fig. 5. Were there occurrences of more than 15% drop of TRW without low values of SPEI (perhaps, in 1973 or 2012)? You may consider marking of such occurrences and of low SPEI in 1963 and 1991, and providing your thoughts on reasons why dry conditions did not caused stress in 1963 and 1991 (looking on monthly climatic data can be useful here, e.g., if conditions were extremely dry in winter, or summer months were dry but cool, etc.), and if there is any available information about not drought-related disturbances in the study area suppressing growth in other years, like pest outbreaks.
L274-275. The meaning of “the warm-dry to warm-wet transition” is not clear.
L298-299. The meaning of term “atmospheric coupling” is not clear.
References. Please, check formatting attentively. There are many instances of incorrect formatting, like species Latin names or book titles being not in italics, or incorrect usage of uppercase letters (IPSS abbreviation in [5], journal title in [15,48], “Pinus” in [30], “Tianshan” in [33], extra capital letters in paper title in [39]). In [6], it should be “Williams, A.P.” as in [7]. In [51], check author names’ list (it’s incomplete, and “Barbara” is the first name) and overall structure of reference. In [59, 61, 64], journal title is not abbreviated. Please check if references [60] and [63] are on different editions of the same book, and if one reference can be used in both cases. In [67,68] format of author list is incorrect.
Reviewer 5 Report
Overall, I found this paper to be a sound scientific contribution to the dendrology literature. Below I point out a few suggestions that will help readers of this article.
Table 1: I did not see methodology to describe the parameters listed in this table.
Methods: Could you add text to describe how you calculate SPEI and SPEI 12.6
Methods: You should add text describing your statistical processes.
Figure 3 caption: Can you add text to help reader quickly understand that P = previous, C = current and the # relates to month. [same with Figure 4]
Figure 6: This figure could be made in a way that makes table 2 not needed. You could add error bars to the triangles and add statistical letters to denote differences between STD groups.
Discussion:
Line 272: What evidence supports the lack of water availability?
Line 314-329: While reading through this I kept wondering about density effects and its effect on drought stress. Are the age groups correlated with stand density? Meaning young trees grow at greater density, then by middle age, stem exclusion has killed off many trees and stem density is less. Any text you could provide to put stand density by age group into context would help the reader.